# BOUNDARY GRAPH NEURAL NETWORKS FOR 3D SIMULATIONS

## ABSTRACT

The abundance of data has given machine learning considerable momentum in natural sciences and engineering. However, the modeling of simulated physical processes remains difficult. A key problem is the correct handling of geometric boundaries. While triangularized geometric boundaries are very common in engineering applications, they are notoriously difficult to model by machine learning approaches due to their heterogeneity with respect to size and orientation. In this work, we introduce Boundary Graph Neural Networks (BGNNs), which dynamically modify graph structures to address boundary conditions. Boundary graph structures are constructed via modifying edges, augmenting node features, and dynamically inserting virtual nodes. The new BGNNs are tested on complex 3D granular flow processes of hoppers and rotating drums which are standard components of industrial machinery. Using precise simulations that are obtained by an expensive and complex discrete element method, BGNNs are evaluated in terms of computational efficiency as well as prediction accuracy of particle flows and mixing entropies. Even if complex boundaries are present, BGNNs are able to accurately reproduce 3D granular flows within simulation uncertainties over hundreds of thousands of simulation timesteps, and most notably particles completely stay within the geometric objects without using handcrafted conditions or restrictions.

## 1 INTRODUCTION

Deep learning (Krizhevsky et al., 2012) dramatically changed scientific fields such as computer vision, natural language processing, or the medical sciences. More recently, deep learning research has been expanded towards physical simulations such as fluid dynamics, deformable materials, or aerodynamics (Li et al., 2018; Ummenhofer et al., 2019; Sanchez-Gonzalez et al., 2020; Pfaff et al., 2020). The progress of deep learning in physical simulations was often driven by Graph Neural Networks (GNNs) (Scarselli et al., 2009; Defferrard et al., 2016; Kipf & Welling, 2017), which proved effective when modeling interactions between many entities via forward dynamics (Battaglia et al., 2018).

Here we want to focus on learning practically relevant granular flow simulations. Granular flows are ubiquitous in nature and industrial processes. Pharmaceutical powders, plastic granulates, or rocks obtained by mining are just some examples of granular media that are used in industries and which are processed in a multitude of different flow states. Consequently, simulations of granular flow processes are required for the design of many industrial processes. Such simulations allow the optimization of devices and machinery in which particle flow is essential. These simulations are needed for a wide range of materials and material mixtures, which can have different cohesion or friction properties. The utilized machines constitute the boundary conditions for the considered granular flow processes. Many conventional simulation approaches, that are based on a solid mathematical theory and which take complex boundaries into account, model these boundaries by triangularizations. Therefore, industrial machinery is often represented by triangular meshes, which can be considered a standard geometric description in the engineering field. Figure 1 exemplarily visualizes triangularized surfaces for two standard components in industrial setups, namely a hopper (left) and a rotating drum (right).

In this work, we try to tackle the problem of accurately modeling granular processes when triangularized boundary surfaces are present. We therefore suggest a simple extension of conventional GNNs: Boundary Graph Neural Networks (BGNNs). BGNNs dynamically modify graph structures

to model interactions of particles and triangulated boundaries. This is achieved by inserting virtual nodes, adding edges and modifying node and edge attributes.

We test the effectiveness of BGNNs on complex 3D granular flow simulations of hoppers and rotating drums. The data for BGNN training is obtained by precise but potentially time-consuming simulations. The accuracy of our BGNN models is measured via various aggregate quantities, namely averaged particle positions, flows, and mixing entropies. We ensure that our BGNN models have high simulation quality by requiring high accuracy of the models in terms of these aggregate quantities. BGNN predictions have the potential to be considerably faster than traditional granular flow simulation methods while keeping the same precision. Code, training and test data will be made public upon publication.

The main contributions of this work are:

- We introduce Boundary Graph Neural Networks (BGNNs), which dynamically modify graph structures via boundary graph structures to allow an accurate modeling of particle interactions with triangularized boundaries.
- We implement BGNNs for learning complex 3D granular flow simulations of hoppers and rotating drums.
- We assess the BGNN simulation quality via the difference of relevant physical quantities between model predictions and simulations. We show that BGNNs are able to generalize granular flow dynamics over hundreds of thousands of timesteps while having the potential to be considerably faster than state-of-the-art simulation methods.

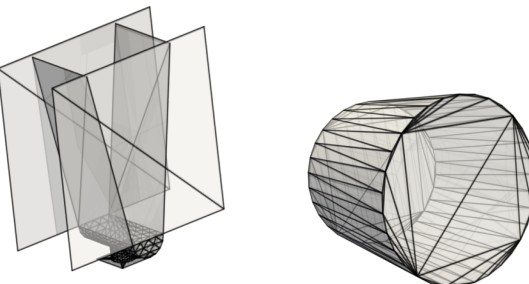

Figure 1: Triangularized boundary surfaces of the hopper (left) and the rotating drum (right). The accurate description of rather simple, curved geometries requires a relatively large number of triangles of different shapes and sizes.

## 2 BACKGROUND

**Dataset.** Due to the fact that there are no governing equations for granular flow like the Navier-Stokes equations for fluid flow (Faccanoni & Mangeney, 2013), a method similar to molecular dynamics, the Discrete Element Method (DEM) (Cundall & Strack, 1979) is used for simulating granular flow. The key idea of DEM is to represent granular media by discrete particles (e.g. spheres or polyhedra), which interact by exchanging momentum using established contact models. LIGGGHTS (Kloss et al., 2012, see App. A) is an open-source DEM implementation, which is able to simulate particle flow for a wide range of materials and complex mesh-based wall geometries and, which therefore enables the simulation of many industrial processes. Consequently, we used LIGGGHTS to generate training, validation and test set simulation trajectories for granular flow within different hopper and rotating drum environments.

**Time transition model.** We build our method upon Sanchez-Gonzalez et al. (2020) and use the semi-implicit Euler method to numerically integrate the equations of motion using model-predicted acceleration. The time-transition from time $t$ to time $t + 1$ is given by $\dot{x}^{t+1} = \dot{x}^t + \Delta t \ \ddot{x}^t$ and $x^{t+1} = x^t + \Delta t \ \dot{x}^{t+1}$, where $x$ is the particle location, and $\dot{x}$ the particle velocity. To calculate the time-transition $x^{t+1}$ the particle acceleration $\ddot{x}^t$ is predicted.

**Graph Neural Networks.**   We consider graphs $\mathcal{G} = (\mathcal{V}, \mathcal{E})$, with nodes $v_i \in \mathcal{V}$ and edges $e_{ij} \in \mathcal{E}$, where $N$-dimensional node features $\boldsymbol{p}_{v_i} \in \mathbb{R}^N$ are attached to each of the nodes. Whether the graph $\mathcal{G}$ contains an edge between a pair of nodes $(v_i, v_j)$ depends on the distance between the nodes:

$$e_{ij} \in \mathcal{E} \iff d(v_i, v_j) \leqslant \text{cut-off}, \tag{1}$$

where the cut-off radius is usually a hyperparamter of the model. Edges might have $M$-dimensional edge features $\boldsymbol{a}_{ij} \in \mathbb{R}^M$ attached to each edge $e_{ij}$. Graph networks are designed to learn from graph-structured data (Scarselli et al., 2009; Kipf & Welling, 2017; Defferrard et al., 2016; Battaglia et al., 2018). Message passing networks (Gilmer et al., 2017) are a specific type of graph neural networks and usually consist of three different types of layers: node and edge feature embedding layers, the core message passing layers, and read-out layers. Message passing iteratively updates the embeddings of edges $(\boldsymbol{m}_{ij})$ and nodes $(\boldsymbol{h}_i)$, i.e., the embeddings of $\boldsymbol{a}_{ij}$ and $\boldsymbol{p}_{v_i}$, at edge $e_{ij}$ and node $v_i$ via:

$$\boldsymbol{m}'_{ij} = \phi(\boldsymbol{h}_i, \boldsymbol{h}_j, \boldsymbol{m}_{ij}), \qquad \boldsymbol{h}'_i = \psi\left(\boldsymbol{h}_i, \square_{e_{ij} \in \mathcal{E}}\, \boldsymbol{m}'_{ij}\right), \tag{2}$$

where the aggregation $\square_{e_{ij} \in \mathcal{E}}$ at node $v_i$ in Eq. (2) is across all nodes that are connected to node $v_i$ via an edge $e_{ij}$. Typically $\square$ represents a mean or max operation. The learnable functions $\phi$ and $\psi$ are commonly presented by Multilayer Perceptrons (MLPs). Equation (2) describes in a compact form the computation and aggregation of messages, and the subsequent update of node embeddings. The final node embeddings are used for predictions via read-out layers.

## 3   BOUNDARY GRAPH NEURAL NETWORKS

**Boundary Graph Neural Networks.**   We introduce Boundary Graph Neural Networks (BGNNs) for modeling the time transition dynamics in simulations within complex geometries. In BGNNs, each graph node $v_i$ is associated to a particle with location $\boldsymbol{x}_{v_i}$, velocity $\dot{\boldsymbol{x}}_{v_i}$ and acceleration $\ddot{\boldsymbol{x}}_{v_i}$, which is similar to Sanchez-Gonzalez et al. (2020). BGNNs modify and enhance the graph structure to include boundaries (see Fig. 2). BGNNs dynamically add $\tilde{n}$ virtual nodes $\tilde{v}_j \in \tilde{\mathcal{V}}$ for boundary regions, iff the corresponding boundary region is within a cut-off radius to any other particle. We augment the set of edges $\mathcal{E}$ by boundary edges $\tilde{e}_{ij}$ giving an enhanced edge set $\hat{\mathcal{E}}$ with $e_{ij} \in \mathcal{E}$ and $\tilde{e}_{ij} \in \tilde{\mathcal{E}}$. Analogously to Eq. (1), the existence of particle-particle edges $e_{ij}$ and particle-boundary edges $\tilde{e}_{ij}$ is determined via:

$$e_{ij} \in \mathcal{E} \subseteq \hat{\mathcal{E}} \iff d(v_i, v_j) \leqslant \text{cut-off}_e, \tag{3}$$
$$\tilde{e}_{ij} \in \tilde{\mathcal{E}} \subseteq \hat{\mathcal{E}} \iff \tilde{d}(v_i, \tilde{v}_j) \leqslant \text{cut-off}_{\tilde{e}}. \tag{4}$$

Note, that the cut-off radii cut-off$_e$ and cut-off$_{\tilde{e}}$ are not necessarily the same, and, $d : \mathcal{V} \times \mathcal{V} \to \mathbb{R}$, while $\tilde{d} : \mathcal{V} \times \tilde{\mathcal{V}} \to \mathbb{R}$, i.e. bidirectional edges are used between real nodes and unidirectional edges are used between real and virtual nodes.

In order to include more information about boundary surfaces into particle-boundary interactions, $\tilde{N}$-dimensional node features that encode information about the inclination of triangles in space are concatenated with the existing node features $\boldsymbol{p}_{v_i} \in \mathbb{R}^N$. Additionally, coordinate information is used both for existing nodes ($\boldsymbol{X} = \{\boldsymbol{x}_{v_0}, \ldots, \boldsymbol{x}_{v_{n-1}}\}$) as well as for virtual nodes ($\tilde{\boldsymbol{X}} = \{\tilde{\boldsymbol{x}}_{\tilde{v}_0}, \ldots, \tilde{\boldsymbol{x}}_{\tilde{v}_{\tilde{n}-1}}\}$). For virtual nodes, the additional coordinates $\tilde{\boldsymbol{x}}_{\tilde{v}_j}$ are chosen such that they minimize the distance between points from boundaries and real particles. The resulting set of node features $\hat{\boldsymbol{P}}$ and node coordinates $\hat{\boldsymbol{X}}$ are:

$$\hat{\boldsymbol{P}} = \{\boldsymbol{p}_{v_0}, \ldots, \boldsymbol{p}_{v_{n-1}}, \tilde{\boldsymbol{p}}_{\tilde{v}_0}, \ldots, \tilde{\boldsymbol{p}}_{\tilde{v}_{\tilde{n}-1}}\}, \qquad \hat{\boldsymbol{X}} = \{\boldsymbol{x}_{v_0}, \ldots, \boldsymbol{x}_{v_{n-1}}, \tilde{\boldsymbol{x}}_{\tilde{v}_0}, \ldots, \tilde{\boldsymbol{x}}_{\tilde{v}_{\tilde{n}-1}}\}, \tag{5}$$

where $\hat{\boldsymbol{p}}_i \in \mathbb{R}^{N+\tilde{N}}$ and $\hat{\boldsymbol{x}}_i \in \mathbb{R}^3$ denote the elements of $\hat{\boldsymbol{P}}$ and $\hat{\boldsymbol{X}}$, respectively. Similarly to above, message passing updates the embeddings of edges $(\hat{\boldsymbol{m}}_{ij})$ and the embeddings of nodes $(\hat{\boldsymbol{h}}_i)$ via

$$\hat{\boldsymbol{m}}'_{ij} = \hat{\phi}\left(\hat{\boldsymbol{h}}_i, \hat{\boldsymbol{h}}_j, \hat{\boldsymbol{m}}_{ij}\right), \qquad \hat{\boldsymbol{h}}'_i = \hat{\psi}\left(\hat{\boldsymbol{h}}_i, \square_{\hat{e}_{ij} \in \hat{\mathcal{E}}}\, \hat{\boldsymbol{m}}'_{ij}\right), \tag{6}$$

where the aggregation $\square_{\hat{e}_{ij} \in \hat{\mathcal{E}}}$ at node $v_i$ in Eq. (6) is across all real or virtual nodes that are connected to $v_i$ via an edge $\hat{e}_{ij}$. Similar to Gilmer et al. (2017) and Satorras et al. (2021), we make use of

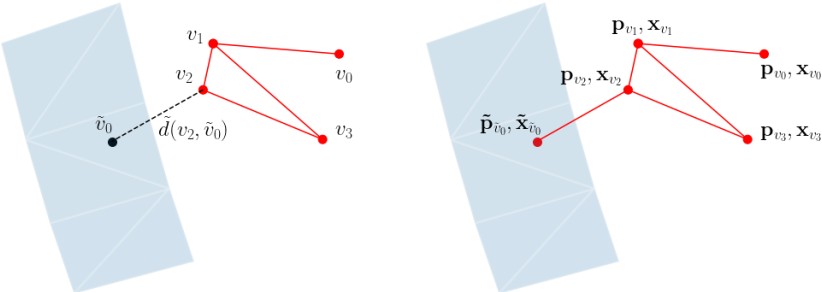

Figure 2: Dynamic modification of the graph edges (red lines) and nodes (red points).Left: Calculation of the distance $\tilde{d}(v_2, \tilde{v}_0)$ between a real particle at node $v_2$ and the triangle corresponding to virtual particle node $\tilde{v}_0$. Right: Insertion of an additional edge between $\tilde{v}_0$ and $v_2$ and representation of the nodes in terms of the corresponding node features $\mathbf{p}_{v_i}, \mathbf{x}_{v_i}$ and $\tilde{\mathbf{p}}_{\tilde{v}_j}, \tilde{\mathbf{x}}_{\tilde{v}_j}$ for real and virtual node features, respectively.

pairwise distances ($\left\|\hat{x}_i - \hat{x}_j\right\|^2$ and $\hat{x}_i - \hat{x}_j$ and deterministic functions thereof). These are for BGNNs between real and between real and virtual particles and we pass this information to the graph network as edge attributes $\hat{a}_{ij}$, for which an initial edge embedding $\hat{m}_{ij}$ is determined via an edge embedding layer. The final node embeddings are used for the predictions via the read-out layers. For aggregation $\square$, we use the mean. The main challenges of the implementation of BGNNs are the modification of the graph structure towards a **dynamic boundary graph structure** to include boundary information via the introduction of modified edges $\tilde{e}_{ij}$, modified node features $\hat{p}_i$, and coordinates for virtual nodes $\tilde{x}_{\tilde{v}_i}$. In the following section, we discuss how $\tilde{e}_{ij}$, $\hat{p}_i$, and $\tilde{x}_{\tilde{v}_i}$ are calculated.

## 4 DYNAMIC BOUNDARY GRAPH STRUCTURE

A boundary graph structure, which is a dynamic modification of the graph structure is needed for two reasons: (i) A static graph structure, which inserts many particles (number proportional to the surface area) for every boundary surface, may result in large computational costs in 3D scenes. (ii) For certain time frames, only some parts of the mesh might be relevant and computations can be saved. For example, as long as particles are in free fall in a container far away from the bottom, the mesh part describing the bottom is irrelevant for the next time step.

**One virtual particle is enough to describe particle boundary interactions.** Every particle "sees" at most one virtual particle representing the boundary surface area, namely that particle which has the shortest distance. Since interaction strength decreases continuously with the distance, it is ensured that for every particle-boundary interaction that boundary point with the largest contribution is considered. Table 1 shows average numbers of nodes $|\mathcal{V}|$, as well as average numbers of boundary edges $|\tilde{\mathcal{E}}|$ and the relative increase in edges (ratio of the number of added wall edges to the total number of particle edges). The scalability of BGNNs would suffer if more than one particle per particle-boundary interaction surface was considered.

Table 1: Growth of the number of edges due to boundaries in the graph. The table shows statistics across the training trajectories of non-cohesive particles in a standard setting in hopper and drum experiments. For each trajectory the frame with maximum relative increase in the number of edges due to virtual particles $|\tilde{\mathcal{E}}|/|\mathcal{E}|$ has been selected as a representative frame. This is done since we are interested in the maximum effect additional virtual particles have on the memory requirements. Number of particles $|\mathcal{V}|$, number of additional virtual edges $|\tilde{\mathcal{E}}|$, and % increase are listed.

| Experiment | $|\mathcal{V}|$ | $|\tilde{\mathcal{E}}|$ | % increase |
|---|---|---|---|
| Hopper | 1113 ± 738 | 5475 ± 3547 | 72.2 |
| Drum | 3283 ± 282 | 1678 ± 188 | 54.8 |

A dynamic boundary graph structure is obtained in three stages:

- Calculating distances between real particles and triangular boundary surface areas in order to decide if a virtual particle node needs to be inserted into the graph, and subsequently obtain additional edges $\tilde{e}_{ij}$
- Obtaining positional coordinates for virtual nodes $\tilde{x}_{\tilde{v}_j}$ as representatives of the relevant fraction of the triangular surface area $\tilde{x}_{\tilde{v}_j}$
- Modifying node features $\tilde{p}_i$ by including normal directions to facilitate the learning of geometric relationships between particles and boundary surfaces

**Modified edges $\tilde{e}_{ij}$ via calculation of distances to boundaries.** To obtain modified edges $\tilde{e}_{ij}$, boundary particles are dynamically inserted, but only if a real particle is close to the corresponding boundary. The insertion of unnecessary edges into the graph is avoided. Such edges would connect nodes of real and virtual particles, although they are far apart. To decide whether a virtual particle has to be inserted requires the calculation of distances between pairs of real particles and mesh triangles. Specifically, the squared distance between the particle center and the closest point on the mesh triangles is calculated (adopted from (Eberly, 1999)). For this purpose, a location on a triangle $t$ is parametrized by two scalar values $u$ and $v \in \mathbb{R}$:

$$t(u, v) = b + u\,e_0 + v\,e_1 \,,$$

where $u \geq 0$, $v \geq 0$, and $u + v \leqslant 1$, $b$ represents one of the nodes of the triangle, and, $e_0$ and $e_1$ are vectors from $b$ towards the other two nodes (see Fig. 3). The minimal Euclidean squared distance $q$ of the point $p$ to the triangle is given by the optimization problem:

$$d = \min_{u,v}\; q(u, v) = \|t(u, v) - p\|^2 \tag{7}$$

$$\text{s.t.}\quad u \geq 0\,,\quad v \geq 0\,,\quad u + v \leqslant 1\,.$$

The minimizing arguments $u'$ and $v'$ parametrize the closest point $t(u', v')$ of the triangle to the point $p$. As indicated in Fig. 3, seven cases have to be to distinguished: one case (c0) in which $t(u', v')$ is located within the (closed) triangle, three cases (c1, c3, c5) in which $t(u', v')$ is located on one edge of the triangle (including the edge corner points as special cases), and three cases (c2, c4, c6) in which $t(u', v')$ is located on one of two edges (including the triangle corner points as special cases). Whether a virtual particle is inserted is determined by Eq. (4) and the particle-triangle distance $d$.

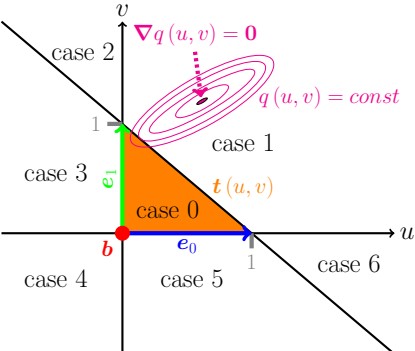

Figure 3: Visualization of point - triangle distance calculations in 3D. The triangle is represented by a parameterized function $t(u, v) = b + u\,e_0 + v\,e_1$ with $u \geq 0$, $v \geq 0$, $u + v \leqslant 1$ (indicated by the orange area). Level sets of $q(u, v)$ are indicated by ellipses and describe the squared Euclidean distance of a triangle point $t(u, v)$ to the point $p$, for which we compute the minimum distance.

**Positional coordinates $\tilde{x}_{\tilde{v}_j}$ for virtual nodes.** After having modified the edges $\tilde{e}_{ij}$ via the calculation of distances between real particles and triangular boundary surface areas, we now have to decide where to place the virtual particles on these boundary surface areas. For virtual particles the coordinates $\tilde{x}_{\tilde{v}_j}$ correspond to $t(u', v')$, i.e., the closest points on the triangular boundary surface areas with respect to the interacting real particle.

**Modifying node features $\hat{p}_i$ by including boundary normal directions.** Typical granular flow simulations comprise substantially more particle - particle interactions than particle - boundary interactions, which may impede the learning of particle - boundary interactions. In Kipf et al. (2018) the problem of qualitatively different interactions is addressed by introducing a dedicated message generating network for each interaction type. We avoid such extensions of our model by means of the following two approaches. First, we introduce additional node features, such that the neural network is able to distinguish the different types of nodes. Second, we adapt the weight initialization of the node feature embedding $\hat{\psi}$, such that the embedding network can be trained with larger values for the additional features. Consequently, the network can learn different dynamics for particle - particle and particle - boundary interactions. The additional node features are: (i) type feature, i.e., a binary indicator of whether a node represents a particle that is real or virtual, and, in the latter case, (ii) the components of the normal vector (see App. B for more information on an orientation-independent representation of the normal vectors) of the triangular surface areas (null vectors for real particles).

## 5  RELATED WORK

There is a rich body of literature on applications of Deep Learning in the context of physics simulations. Most notably related to BGNNs are the works of Sanchez-Gonzalez et al. (2020), Ummenhofer et al. (2019), and, Li et al. (2018), all of which propose methods of learning particle simulations without enforcing constraints. These approaches can be contrasted to works like Ladickỳ et al. (2015) or Schenck & Fox (2018) that utilize strong inductive biases. Ladickỳ et al. (2015) construct features for Random Forest Regression that are influenced by Smooth Particle Hydrodynamics (Gingold & Monaghan, 1977; Lucy, 1977). Schenck & Fox (2018) construct a differentiable fluid dynamics network that is closely related to the Position Based Fluids method (Macklin & Müller, 2013). Importantly, both methods are built on the assumption that the governing equations of the system are known, which is typically not the case for granular flow dynamics.

Complex mesh-based wall geometries have been employed to compute updates for nodes of the mesh itself (Pfaff et al., 2020). In contrast, our simulator utilizes the mesh to represent static boundaries in a highly efficient way. We share the opinion of Sanchez-Gonzalez et al. (2020) that the network architecture with continuous convolutions as suggested by Ummenhofer et al. (2019) can be interpreted as GNNs. In doing so, a difference to Sanchez-Gonzalez et al. (2020) and our work is that Ummenhofer et al. (2019) use static particles as special nodes in the first message passing step only. Consequently, the framework of Sanchez-Gonzalez et al. (2020), which is based on Battaglia et al. (2018), appears to be the most general to us, performing well even without explicit hierarchical clustering as suggested in DPI-Net (Li et al., 2018). Experiments of Sanchez-Gonzalez et al. (2020) further suggest that their simulation of sand particles are superior to the implementation of Ummenhofer et al. (2019). However, Sanchez-Gonzalez et al. (2020) only consider simple cuboid boundaries for their 3D simulations, leaving more realistic complex geometries as an open and yet untouched challenge. Furthermore, they use sampled, static particles to represent boundaries for 2D simulations, which in general does not scale well for 3D simulations due to the quadratic increase of boundary particles (square areas instead of lines).

## 6  EXPERIMENTS

We test the effectiveness of BGNNs on complex 3D granular flow simulations. The development, design, and construction of many mechanical devices is based on granular flow simulations. These devices can have very different geometries and must be designed for a wide range of materials with highly varying properties. For example, cohesion properties can range from dry, wet, to oily. In the simulations, we consider very common device geometries and different cohesion properties to cover a wide range of situations with our available computational resources. The two common geometries are hoppers and rotating drums (see Fig. 1, Fig. 4, and Fig. 5). The two different cohesion properties are non-cohesive describing liquid-like, oily materials and cohesive describing dry, sand-like materials. We compare the BGNN predictions to the simulations in two aspects: speed and accuracy.

**Simulation Details.** For all experiments, gravitation acts along the $z$-direction. The upper part of the hopper is delimited along the $y$-axis by two planes, which are parallel to the $x$-$z$ plane (see Fig. 4). The $x$-axis is delimited by two planes, that are inclined at certain angles $\alpha$, $180° - \alpha$ to the

$x$-$y$ plane and at corresponding angles $\alpha - 90°, 90° - \alpha$ to the $y$-$z$ plane. The hopper has an initially closed hole at the bottom, which has an adjustable radius. The rotation axis of the drum is the $y$-axis (see Fig. 5). The initial filling of the hopper and drum is done by randomly inserting particles into a predefined region. More information can be found in App. C. We use around 1000 and around 3000 particles for hopper and rotating drum simulations, respectively. In order to have trajectories with non-cohesive and cohesive particles, we use the simplified JKR model (Roessler & Katterfeld, 2019) with a cohesion energy density of 0 J/$m^3$ and $10^5$ J/$m^3$ for non-cohesive and cohesive particles. The training data consists of 30 simulation trajectories, where each trajectory consists of 100.000 (250.000) simulation timesteps for hopper (rotating drum). For BGNN training every 40 (100)-th timestep is used. Trajectories have different angles $\alpha$ and different hole radii (hopper) and different initial particle placement (drum). Moreover, the number of particles is varied by ±25%.

**Implementation Details.** We use 5 message passing layers, with 128 and 512 nodes for intermediate node and edge representation. The cut-off radii strongly depend on the particle size. We use cut-off radii of 0.02 and 0.008 for rotating drum and hopper, respectively. Cut-off radii have been treated as hyperparameter of our model. More details can be found in App. C.

**Assessment Of Physical Quantities.** Granular flow simulations should correctly describe systems on macroscopic scales in terms of **particle-averaged positions** $\bar{\mathbf{x}}(t)$ and **particle flows** $\bar{\mathbf{v}}(t)$ for $n$ particles as a function of time: $\bar{\mathbf{x}}(t) = \frac{1}{n} \sum_i \mathbf{x}_i(t)$ and $\bar{\mathbf{v}}(t) = \frac{1}{n} \sum_i \mathbf{v}_i(t)$. Hoppers are devices that aim at adjusting the flow of particles along the direction of gravity, which coincides with the $z$-axis in our experiments. Rotating drums are commonly utilized as mixing devices for various applications in e.g. industry, research, and agriculture. They are essentially rotating cylinders that are partially filled with a granular material. The mixing property of these devices is a result of numerous particle interactions under time-varying boundary conditions. For rotating drum experiments, we quantify the extend of particle mixing via the **mixing entropy** (Lai & Fan, 1975). If the z-coordinate of a particle's initial position $\mathbf{x}_i(0)$ is above (below) the median z-coordinate of all particles in the initial state, we assign it to class $c = +1$ ($-1$). Based on this assignment local entropies $s(\boldsymbol{g}_{klm}, t)$ at grid cells $\boldsymbol{g}_{klm}$ are calculated, where the indices $klm$ identify an individual grid cell. The local entropies $s(\boldsymbol{g}_{klm}, t)$ are computed from particle counts $n_c(\boldsymbol{g}_{klm}, t)$, of the respective classes $c = \pm 1$. The total number of particles in a grid cell is obtained by $n(\boldsymbol{g}_{klm}, t) = n_{+1}(\boldsymbol{g}_{klm}, t) + n_{-1}(\boldsymbol{g}_{klm}, t)$. Calculating the particle-number weighted average of the local mixing entropies yields the mixing entropy $\mathrm{S}(t)$ of the entire system:

$$\mathrm{S}(t) = \frac{-1}{\sum\limits_{klm} n(\boldsymbol{g}_{klm}, t)} \sum\limits_{klm} \sum\limits_{c=\pm 1} n(\boldsymbol{g}_{klm}, t) \left( f_c(\boldsymbol{g}_{klm}, t) \log f_c(\boldsymbol{g}_{klm}, t) \right), \tag{8}$$

where $f_c(\boldsymbol{g}_{klm}, t)$ denotes the relative fraction of class $c$ particles in cell $\boldsymbol{g}_{klm}$ at time $t$.

**Results.** In Fig. 4 and Fig. 5 results for the hopper and the rotating drum simulations are presented. The left parts visualize granular flow snapshots at different time steps, both for cohesive and non-cohesive materials. The right parts of the figures include average position and particle flow plots for hopper, as well as particle flow and mixing entropy plots for rotating drum simulations. The simulation uncertainties arise due to the different distributions of the initial filling and due to a ±25% variation in the number of particles across simulations. Short video clips of the trajectories are added as supplement. The difference between cohesive and non-cohesive particles is evident.

BGNNs have learned to model granular flow simulations over thousands of time steps. Most notably, hardly any particle leaves the geometric boundaries. This is achieved without using handcrafted conditions or restrictions on the positions of the particles. Furthermore, BGNNs have learned to model particle-boundary interactions and in doing so correctly represent the dynamics within the system. The predicted quantities are within uncertainties of the simulations. Therefore, we consider the BGNN predictions as sufficiently precise to substitute the simulations. Figure 6 shows out-of-distribution (OOD) scenarios, where the devices are changed with respect to the training data. The hole size of the hopper is decreased in mean by ∼ 50%, while side wall inclination angles have been increased by ∼ 15°. For the drum the length of the corresponding cylinder was increased in mean by ∼ 50%. Our experiments show that our model generalizes well across variations in the geometry. This finding demonstrates that trained BGNNs could be used in the design process of a device to study variations of geometries without retraining the model.

Table 2 gives a run-time comparison of the LIGGGHTS simulation versus a forward pass of BGNNs, which only predict every 100 time steps. The highly optimized CPU algorithm (LIGGGHTS) and a non-optimized GPU compatible algorithm (BGNNs) are compared via their wall-clock times since the hardware settings are quite different. Nevertheless, Tab. 2 shows that the wall-clock time of BGNNs is shorter than the wall-clock time of the simulation. The usage of more particles, would further increase the lead of BGNNs over the simulation in terms of wall-clock time. For the time comparison, we use a typical simulation trajectory from our datasets with 3,408 particles, which needs approximately 2 GB GPU memory for one forward pass. There is potentially even more space for improvement of the BGNN predictions over simulations due to the so called **Young's modulus**. For simulations, it is often assumed that energy is purely transmitted through Rayleigh waves. Thus the time step of DEM simulations is targeted to be a fraction of the propagation time through a single, solid particle. As such the propagation time depends on material parameters, most notably the Young's modulus. However, for several materials the Young's moduli that reflect the true material properties, would lead to extremely small propagation times, which in turn means much more simulation steps. Consequently, much smaller Young's moduli are considered as an approximation, which is valid for gravity driven flows (Coetzee, 2017). However, for many cases, e.g. the penetration of a particle bed by an object, this approximation breaks down (Lommen et al., 2014). BGNNs have the potential to be trained on very small time steps reflecting the true Young's moduli and consequently generalize over much more than "just" 40 or 100 time steps.

Table 2: Runtime comparison for one granular flow process consisting of 250.000 simulation timesteps, which are 2500 BGNN predictions.

| method | device | specification | time steps | wall-clock time [$s$] |
|---|---|---|---|---|
| LIGGGHTS | CPU | AMD EPYC 7H12 | 1 | 356 |
| BGNNs | GPU | NVIDIA A100 | 100 | 158 |

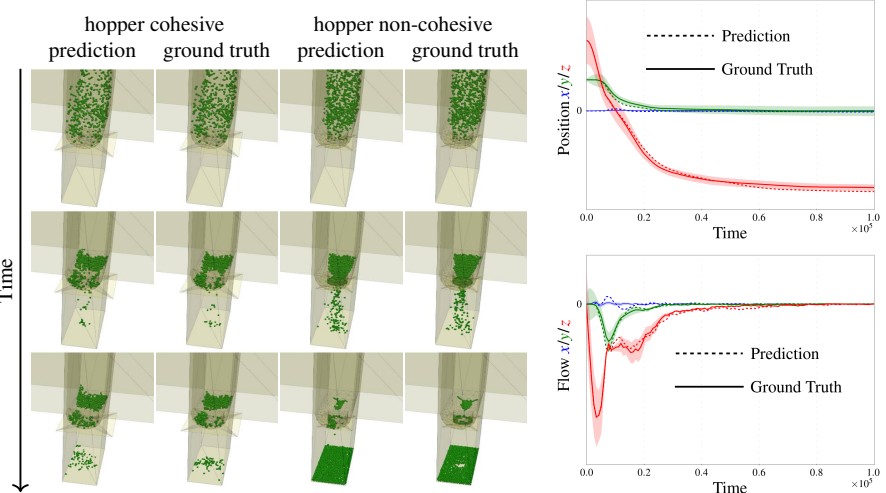

Figure 4: Hopper dynamics. Left: Distributions for cohesive and non-cohesive particles. Simulation data and BGNN predictions are compared. Particles are indicated by green spheres, triangular wall areas are yellow, the edges of these triangles are indicated by grey lines. In contrast to liquid-like non-cohesive particles, cohesive particles lead to congestion of the hopper. Right: Position (upper right) and flow profile (lower right) for non-cohesive particles. Corresponding plots for cohesive particles can be found in App. C. Simulation data (solid lines) and BGNN predictions (dashed lines) are compared. Simulation uncertainties are due to a change of the particle numbers (±25%) and to different initial conditions. To support the reviewing process, we provide simulation predictions for a hopper with more timesteps in animations at `https://bgnn3dsim.bitbucket.io/`.

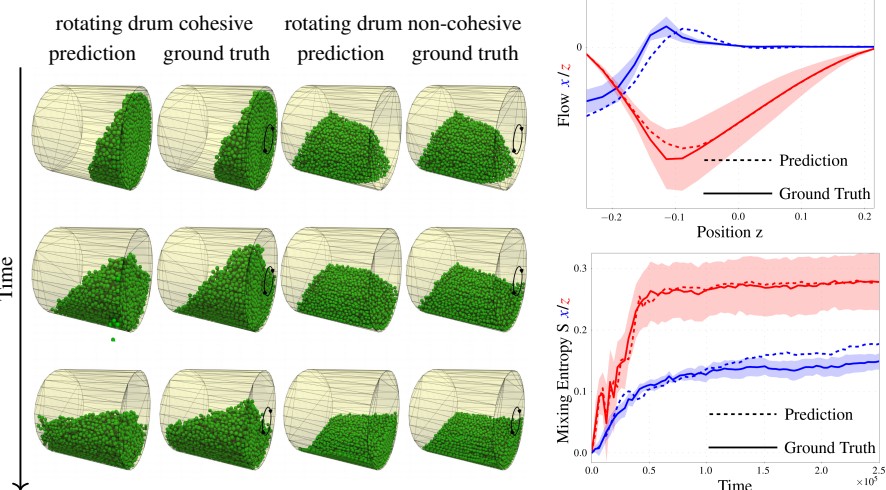

Figure 5: Rotating drum dynamics. Left: Particle distributions for cohesive and non-cohesive particles. Simulation data and BGNN predictions are compared. Particles are indicated by green spheres, triangular wall areas are yellow, the edges of these triangles are indicated by grey lines. The circular arrow indicates the rotation direction of the drum. In contrast to liquid-like non-cohesive particles, cohesive particles stick together much stronger. Right: position (upper right) and entropy plot (lower right) for non-cohesive particles. The entropy is shown for particle class assignment according to the x (blue) and z (red) position. Corresponding plots for cohesive particles can be found in App. C. Simulation data (solid lines) and BGNN predictions (dashed lines) are compared. Simulation uncertainties are due to a change of the particle numbers (±25%) and to different initial conditions. To support the reviewing process, we provide simulation predictions for a rotating drum with more timesteps in animations at https://bgnn3dsim.bitbucket.io/.

## 7 CONCLUSION AND FUTURE DIRECTIONS

We have introduced Boundary Graph Neural Networks (BGNNs) in order to achieve an accurate neural network modeling of simulated physical processes with complex geometries. BGNNs dynamically modify graph structures via modifying edges, augmenting node features, and dynamically inserting virtual nodes. We have tested BGNNs on complex 3D granular flow processes of hoppers and rotating drums, where BGNNs are able to accurately reproduce these flows within simulation uncertainties over hundreds of thousands of timesteps. Most notably particles stay within the geometric objects without using handcrafted conditions or restrictions. However, it should be mentioned that successful simulations often require precise hyperparameter tuning. So far we have not investigated materials with high Young's moduli, and thus we have not yet tested the full generalization properties of BGNNs for such scenarios as described in Sec. 6. Another interesting exten-

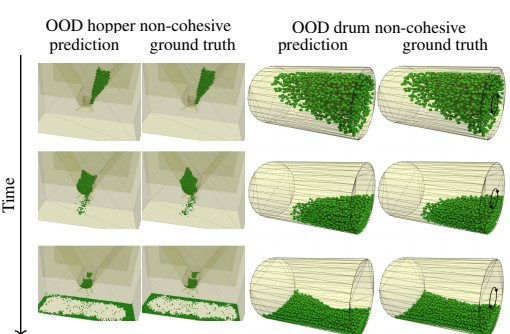

Figure 6: OOD generalization behavior for the hopper (left) and the rotating drum (right). In contrast to the training and validation data the outlet size of the hopper was decreased, the inclination angles of the hopper side walls are enlarged, and, the length of the rotating drum is increased.

sion of our work might be to introduce a velocity dependent cut-off radius, and in doing so to also consider those interactions which are going to happen within the next timesteps although the spatial distance for particles and respective boundaries is still large.

## 8 REPRODUCIBILITY STATEMENT

Nearly all data used in this work is generated by ourselves. The generalization properties strongly depend on the variability of the training samples since we used a limited number of training samples due to computational reasons. We made sure to keep track of the used parameters to be able to reproduce these datasets. To support the reviewing process, we provide simulation predictions for a hopper with more timesteps in animations at `https://bgnn3dsim.bitbucket.io/`

We have included error bars and uncertainty estimates wherever we found it necessary and appropriate. For example, for the hopper and drum example, we considered simulation uncertainties. We selected hyperparameters based on a separate validation set. Besides noise terms for the sack of regularization, one might fairly similar models if applying the same training and hyperparameter selection procedure. We have described our architecture and implementation details in Appendix C. We have further provided supporting concepts and experiments in the appendix. For reproducibility, we plan to provide our code upon acceptance.

For reproducibility, we plan to provide our code upon acceptance.

## 9 ETHICAL STATEMENT

BGNNs might serve as a valuable tool to obtain speedups for the simulation of industrial processes. However, models obtained, may strongly depend on the variability of the training data. Although we performed some OOD experiments, that showed that our models might still perform well under certain geometric aberrations, it is quite clear that a single model will not be able to properly predict the flow dynamics of any arbitrary conceivable granular flow setups without ensuring that this behaviour might have been successfully derived from the limited training data. Therefore, the employment of BGNNs needs careful validation wrt. the specific application in mind.

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

## A  DISCRETE ELEMENT METHOD (DEM) SIMULATOR LIGGGHTS

The open source DEM software LIGGGHTS (Kloss et al., 2012) is based on the Molecular Dynamics code LAMMPS (Plimpton, 1995) developed by Sandia National Labs. Due to the similarity of the underlying algorithms for neighbor list construction, output and parallelism this provided a stable basis for the contact models required for DEM. LIGGGHTS added support for triangular mesh walls, particle insertion and new particle shapes (multispheres and superquadrics). Several of those changes resulted in upstream contributions in LAMMPS.

Over the years LIGGGHTS has become a widely used software in both academia and industry that supports both cutting edge research and industrial applications. Support for several physical phenomena as, e.g. liquid transfer on particles, was instrumental in its success. However, it also highlighted requirements for additional research. In industrial applications there often is the need to study physical phenomena which occur on different time scales, e.g. particle collisions ($O(10^{-5}s)$) vs. moisture content in particles ($O(1s)$, (Mellmann et al., 2011)), which can lead to weeks of simulation time. While advances have been made to overcome such issues (e.g. (Kloss et al., 2017)), they remain limited in their application, due to the fact that they rely on prior simulation of the exact setup and cannot be used for interpolation of quantities directly related to the flow behavior.

In 2019 LIGGGHTS was again forked and forms the basis of the commercial DEM software Aspherix® which expands the capabilities of LIGGGHTS with polyhedral particles, a significantly simplified input language and graphical user interface.

## B  NORMAL VECTOR REPRESENTATIONS

There is an ambiguity in the representation of planes via normal vectors due to the two possible orientations of the normal vectors, which correspond to the same geometry. In general, there are two possibilities of including normal vector information into the model: (i) encoding always that triangle plane normal vector which always points towards or away from the corresponding particle, (ii) including positively and negatively oriented versions of the normal vector, and order them. We decided for option (ii) since pathological cases where the particle is in the same plane as the triangle are avoided and training is further stabilized. In doing so, we have to deal with the fact that the network predictions should be invariant with respect to the orientation of the normal vectors. Therefore, we define a partial ordering which is able to sort the normal vectors with respect to their orientations. For a given normal vector $n = (n_1, n_2, n_3) \in \mathbb{R}^3$, we use the following partial order function

$$f_o(n) = \sum_{i=1}^{3} 3^{i-1} \left( \text{sgn} (n_i) + 1 \right) \tag{B.1}$$

to retrieve the scalar values $f_o(n)$ and $f_o(-n)$ and sort the two vectors according to their corresponding mapped values. Different sign combinations of the normal vectors are shown in App. Fig. B.1. To test the performance of our approach, we conduct a toy experiment as well as a simulation experiment with different representations of normal vectors. We describe both experiments in App. B.1 and App. B.2.

### B.1  REFLECTION TOY EXAMPLE

We conduct a toy experiment to showcase that a partial ordering of normal vectors is helpful for learning 3D simulations. In detail, we consider reflection ($Ref$) at a plane $n$ as given by

$$Ref_n (v) = v - 2 \frac{v \cdot n}{n \cdot n} n , \tag{B.2}$$

and try to learn the reflection formula by a simple ReLU network, which takes the 3 components of $n$ and $v$ as input features and predicts the 3 components of $Ref_n (v)$. The training data consists of reflections at four fixed walls: the top, the bottom, the left, and, the right side of a simple cube.

We use normal vectors of these walls, that point towards the inner of the cube. When evaluating the performance of the trained models, we observe decent predictions, if the orientation of the normal vectors describing the inclination of the walls was equal to the training data (see R1-R4 in Fig. B.2). However, for inverted normal vectors in the test set, only networks which take a partial ordering of the normal vectors into account predict the reflection correctly (see R3, R4 in Fig. B.3).

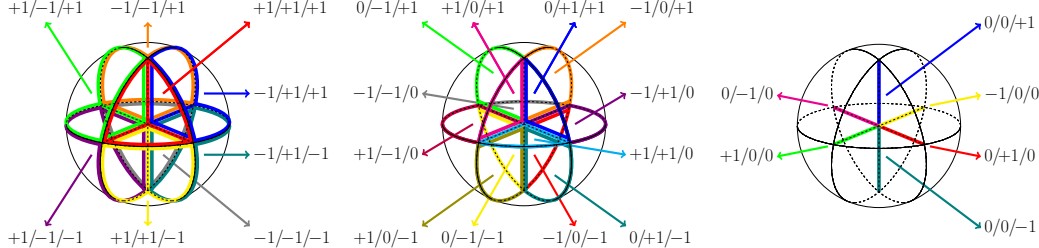

Figure B.1: Partial ordering of normal vectors. The numbers indicate the signs of normal vector components, which are used in the partial order function. The figures from left to right visualize different (ordered) sign combinations. Sets of sign combinations without zero values form volumes (left), sets with one zero value form planar areas (middle), and sets with two zero values form line sections (right). The 0/0/0 combination forms a point at the origin.

## B.2 SIMULATION EXPERIMENT

We compare three different versions of how to include normal vector information for the hopper particle flow experiments:

- not including normal vector information, and filling six node features up with zero entries instead (V1)
- including single normal vector orientation, which is given by the triangle corner point order of the mesh (V2)
- including both normal vector orientations (six features) (V3).

From an information perspective, it should be noted that (i) distance information (scalar distance and distance vectors) to the walls is present in the edge features of the graph and (ii) in most cases the used normal vectors are oriented towards the outside of relevant border walls.

The different particle distribution trajectories obtained by the three versions are compared by computing the Earth Movers distances (Bonneel et al., 2011; Flamary & Courty, 2017, EMD) of predicted and simulated trajectories. We use Euclidean distances for the cost matrix, which we compute at time steps $2^0, 2^1, \cdots, 2^{16}$ for 5 training trajectories and 5 test trajectories. Table B.1 shows the means ($\mu$) and standard deviations ($\sigma$) of EMD values at different time steps and from 5 different training and test trajectories. A paired Wilcoxon test on the concatenated trajectories, shows that V3 significantly outperforms V1 (p-value 2.42e-04) and V2 (p-value 1.50e-03) on the test data.

Interestingly, there is less significance on the training data, which might indicate that the usage of orientation-independent features to represent walls, helps to improve generalization performance, while it might not be that helpful for optimization purposes alone.

Table B.1: Usage of different normal vector information in hopper particle flow experiment. The table summarizes means ($\mu$) and standard deviations ($\sigma$) of the EMD for the different versions and shows the results of a paired Wilcoxon test.

| Version | | Train | | | Test | | |
|---|---|---|---|---|---|---|---|
| | | $\mu$ | $\sigma$ | p-value Row < V3 | $\mu$ | $\sigma$ | p-value Row < V3 |
| V1 | No normal vector | 5.06e-05 | 1.17e-04 | 2.36e-02 | 6.80e-05 | 1.59e-04 | 2.42e-04 |
| V2 | Single normal vector | 1.15e-04 | 3.84e-04 | 3.40e-03 | 1.21e-04 | 4.33e-04 | 1.50e-03 |
| V3 | Both orientations | 5.99e-05 | 1.77e-04 | | 6.36e-05 | 2.06e-04 | |

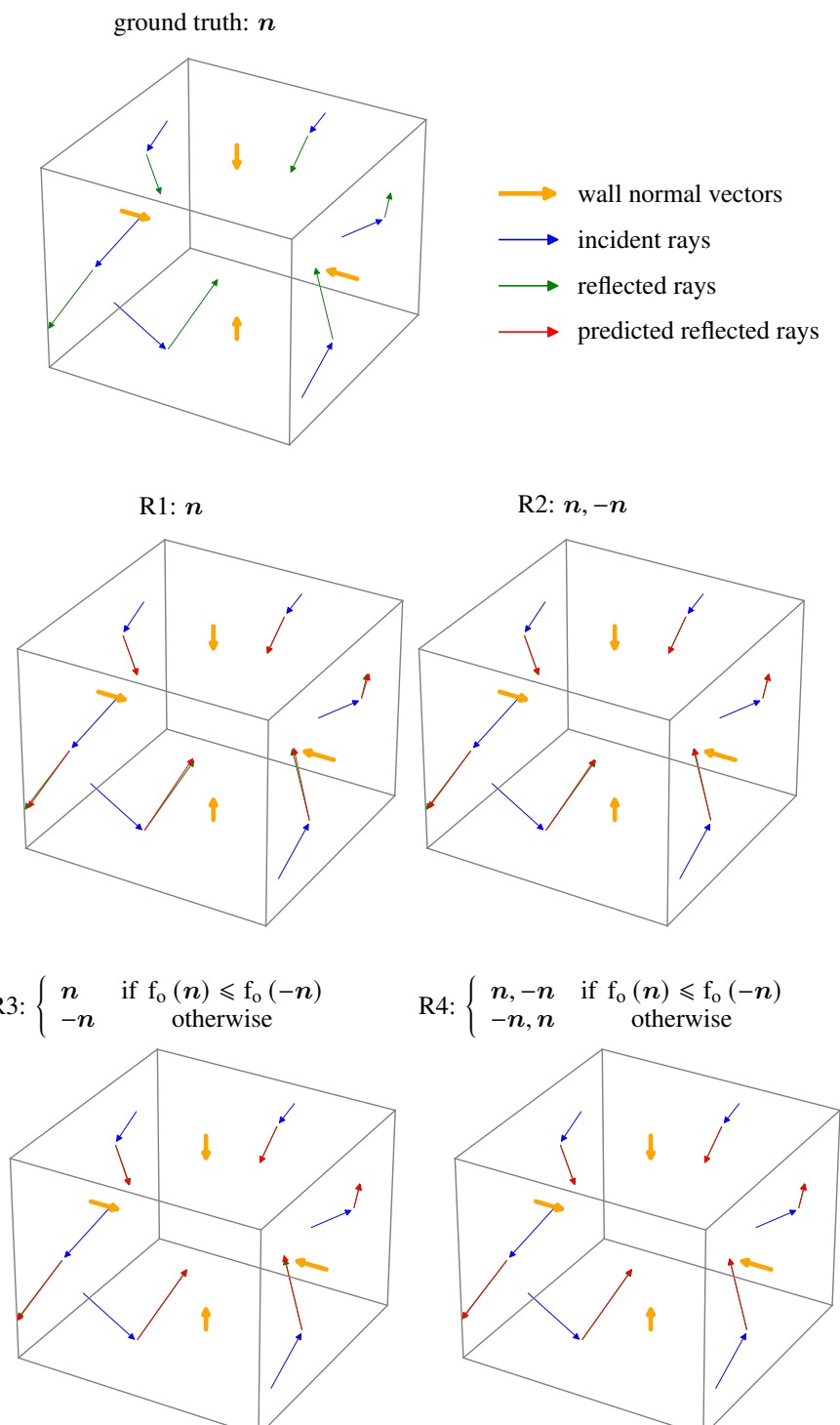

Figure B.2: Reflection of rays at four different walls (left, right, bottom, top). Wall normal vectors are visualized by bold orange arrows. The incident rays are visualized by blue arrows, reflected rays by green arrows, and neural network predictions by red arrows. Neural network predictions are based on wall representations **that are oriented the same way as in the training phase**. The caption above each plot indicates the wall input features used for training each of the networks.

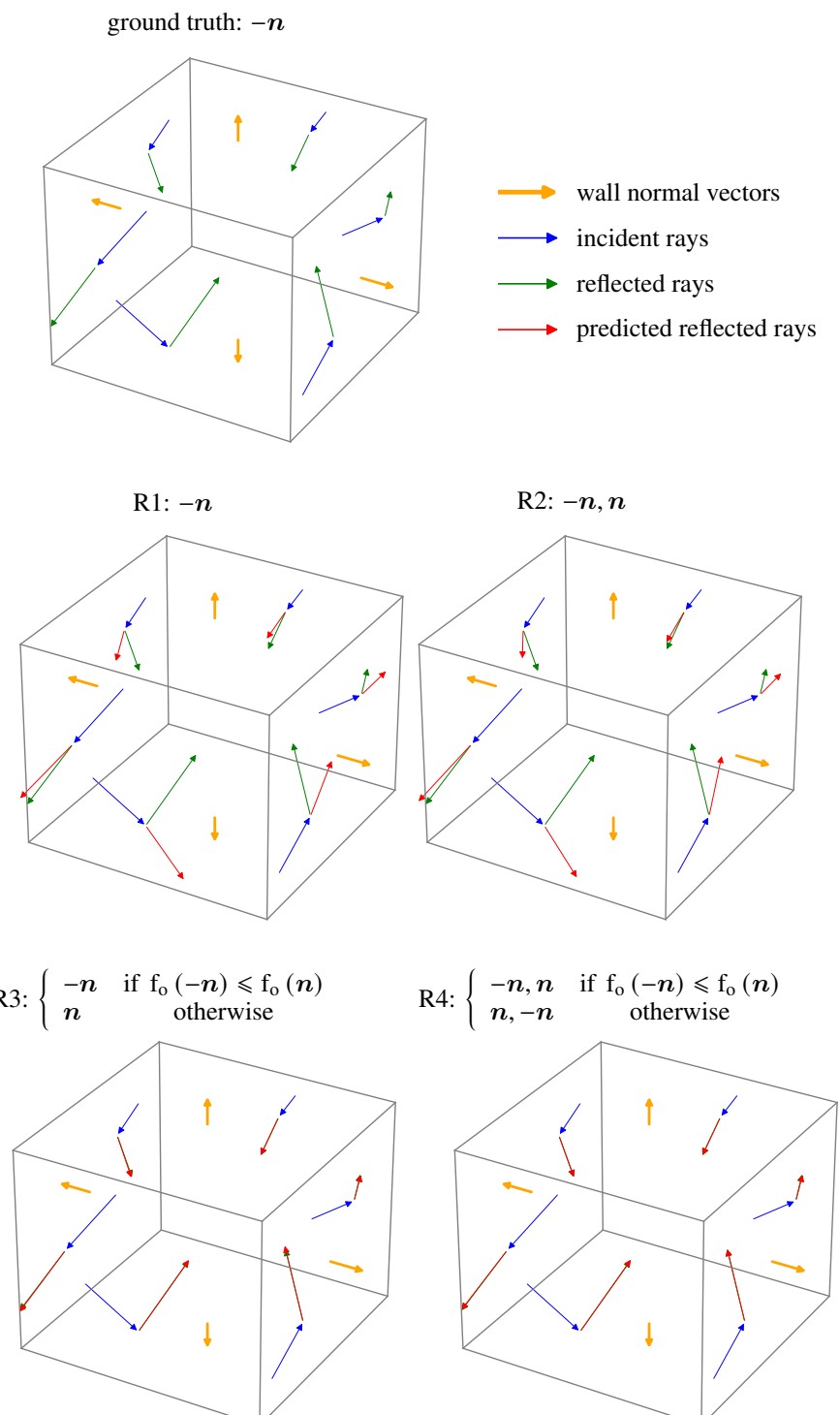

Figure B.3: Reflection of rays at four different walls (left, right, bottom, top). Wall normal vectors are visualized by bold orange arrows. The incident rays are visualized by blue arrows, reflected rays by green arrows, and neural network predictions by red arrows. Neural network predictions are based on wall representations **that are inversely oriented compared to the training phase**. The caption above each plot indicates the wall input features used for training each of the networks.

## C   EXPERIMENTS

### C.1   SIMULATION DETAILS

**Hopper**   The initialization consists of two phases. In a first step, particles are randomly inserted into a small cuboid which is positioned at a certain height above the closed hole of the hopper. This cuboid is continuously filled with particles during the initialization phase and afterwards particles freely move downwards (along the direction of gravity). In this way, the hopper is filled up to a certain height with 20,000 particles. In a second phase, we cut out particles from the filled mass of particles. We do this (i) by applying randomly selected functions and by (ii) randomly filtering out particles from the whole particle mass. The randomly selected functions are e.g. hyperplanes, where we only keep particles if they are at the same side of the hyperplane. The inserted particles have a radius of 0.002 m.

**Drum**   For initialization we assume that the direction of gravity is different than the usual gravitation direction. We insert particles at two random fixed regions within the drum. After the particles are inserted, they can move according to the gravitation direction during the initialization phase. In this way, we obtain different initial particle distributions within the drum. The inserted particles have a radius of 0.01 m.

### C.2   IMPLEMENTATION DETAILS

**Graph Neural Network**   Raw inputs to our graph networks are initial particle positions and the particle positions from the 5 previous frames of the simulations. From these positions velocities are computed. Further inputs include the particle type and the coordinates of the triangle mesh of the respective time frame. We use residual connections (He et al., 2016) for both node and edge updates. For both updates, we use simple two-layer MLP networks, ReLUs (Nair & Hinton, 2010) after the first layer, and layer normalization (Ba et al., 2016) without an additional activation after the second layer. For layer normalization we consider the $\epsilon$-parameter as a hyperparameter and set it to 1.0. The networks for input embedding and read-out are similar to the message passing layers without layer normalization. The network weights are initialized similar to He et al. (2015); for the input embeddings we assume an increased number of input neurons for `fan_in`, where we consider the additional neurons as virtual copies of e.g. the wall indication feature in order to be able to upweight the influence of these features. We use the mean-squared error as an objective and train with Adam optimization (Kingma & Ba, 2015). In order to facilitate learning, we provide as hyperparameter options not only $\left\|\hat{x}_i - \hat{x}_j\right\|^2, \hat{x}_i - \hat{x}_j$ as features to the network, but also $\frac{1}{\left\|\hat{x}_i - \hat{x}_j\right\|}, \frac{\hat{x}_i - \hat{x}_j}{\left\|\hat{x}_i - \hat{x}_j\right\|^2}$ and $\frac{1}{\left\|\hat{x}_i - \hat{x}_j\right\|^2}, \frac{\hat{x}_i - \hat{x}_j}{\left\|\hat{x}_i - \hat{x}_j\right\|^3}$ reflecting the inverse distance law and the inverse-square law, which are present in many physical laws. We normalize input and target vectors and use a variant of Kahan summation (Kahan, 1965; Klein, 2006) in order to compute numerically stable statistics across particles of our dataset.

**Hyperparameter Selection**   We keep 5 trajectories for each setting aside for validation. Criterions for hyperparameter selection are (i) that particles stay within the geometric object, and (ii) that the ground truth trajectory is reproduced.

### C.3   EXPERIMENTAL RESULTS FOR COHESIVE MATERIAL

In the following, experimental results for a cohesive material are shown. The results in the main paper are obtained for non-cohesive granular material, i.e. material with a cohesion energy density of $0 \text{ J}/m^3$, which results in liquid-like behaviour. Increasing the cohesion energy density to $10^5 \text{ J}/m^3$ corresponds to cohesive granular material, i.e. the particles have a strong tendency to clump together. Figure C.1 shows the corresponding comparison of physical quantities for the cohesive granular material. Like in the non-cohesive case, the predictions for cohesive granular material are widely in agreement with the ground truth simulation.

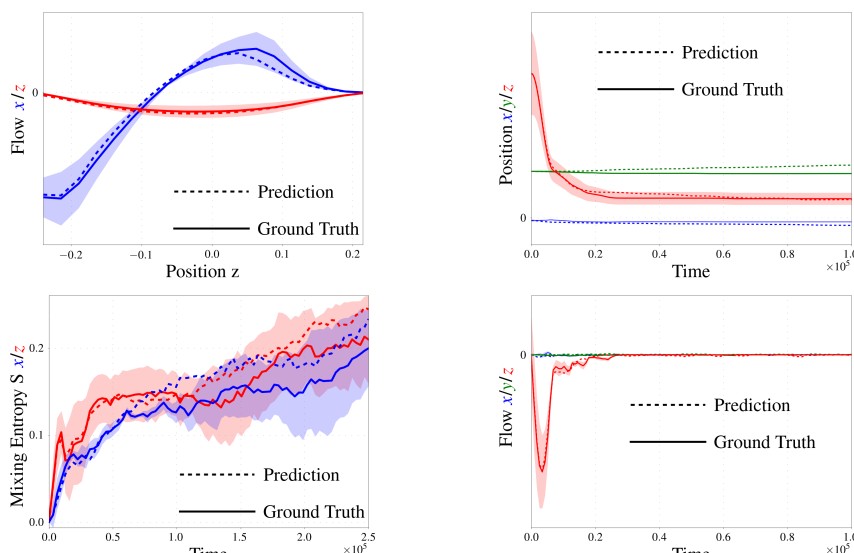

Figure C.1: Experimental results for cohesive granular meterial. Left, top: Time integrated particles flow in rotating drum in the x and z direction as a function of the position along the z axis. Left, bottom: Mixing entropies in the rotating drum as a function of time for particle class assignments according to the x (blue) and z (red) position. Right, top and bottom: Average particle position and particle flows for the hopper as a function of time.

### C.4 A WORD ON HOPPER OOD EXPERIMENTS

For hopper geometries (see e.g. Fig. 4), OOD experiments are characterized by an increase of the side wall inclination angles and by an decrease of the radii of the outlet sizes. Especially due to the latter, we expect fewer particles to hit the ground for OOD architectures if particle-particle and particle-boundary interactions are correctly modeled. In order to statistically test OOD trajectories against in-distribution trajectories, we consider the proportion of particles which have traversed through the outlet of the hopper. We therefore create 15 in-distribution and 15 OOD trajectories for both cohesive and non-cohesive materials. We then apply a Mann-Whitney U test which assesses the proportion of in-distribution against the proportion of OOD particles traversing the outlet. The null hypotheses is that the same or a higher proportion of particles traverses the outlet for the case of an OOD trajectory compared to an in-distribution trajectory.

Table C.1: Comparison of the proportion (mean $\mu$ and std $\sigma$) of particles beyond the outlet of the hopper

| domain | cohesive | | non-cohesive | |
|---|---|---|---|---|
| | $\mu$ | $\sigma$ | $\mu$ | $\sigma$ |
| in-distribution | 0.34 | 0.09 | 0.89 | 0.03 |
| OOD | 0.14 | 0.11 | 0.73 | 0.15 |

The Mann-Whitney U test shows that the predicted proportion values are significantly lower for OOD than for in-distribution trajectories (p-value $< 1.4 * 10^{-4}$ for the cohesive material and p-value $< 1.2 * 10^{-4}$ for the non-cohesive material). We remind the reader that this is expected due to the on average reduced outlet size in OOD geometries. Furthermore, we compare the predicted proportion values of the cohesive and the non-cohesive model under the null hypothesis that the same or a higher proportion of particles traverses the outlet for the case of cohesive trajectories compared to non-cohesive trajectories. The applied Mann-Whitney U test yields a p-value $< 1.70 * 10^{-06}$ for the alternate hypothesis that the proportion is lower for the cohesive model, which is also in agreement with rational arguments (cohesive particles tend to clump together) and observations.

### C.5 MIXER

Additionally to hopper and rotating drum geometries, we apply BGNNs to the geometry of rotating mixers. Interactions within the mixer geometry are especially challenging, since a large number of particles is affected indirectly by the blades at the border and especially those in the middle of the mixer. Figure C.2 shows mixer dynamics The left parts visualize granular flow snapshots at different time steps, where BGNN predictions and ground truth data are contrasted. The right parts of the figure include a flow profile and an entropy curve plot.

### C.6 ABLATION STUDIES

For ablating BGNNs, three design choices are verified:

1. Is our sampling procedure more effective than a dense sampling of the triangularized boundary surface areas?

2. Is a unidirectional particle-wall interaction sufficient to learn the corresponding particle-wall dynamics?

3. Is the mean node aggregation the right choice for the message passing node updates as outlined in Eq. (2).

In Figs. C.3, C.4, C.5, C.6 these three points are assessed for cohesive and non-cohesive particles in hopper and rotating drum geometries, respectively. Consistent results across both geometries and particle types are: (i) dense sampling of the boundary surface areas did not learn particle - wall interactions well, and, (ii) max node aggregation did not seem to work. For including bidirectional edges instead of unidirectional ones, the behaviour was not completely consistent.

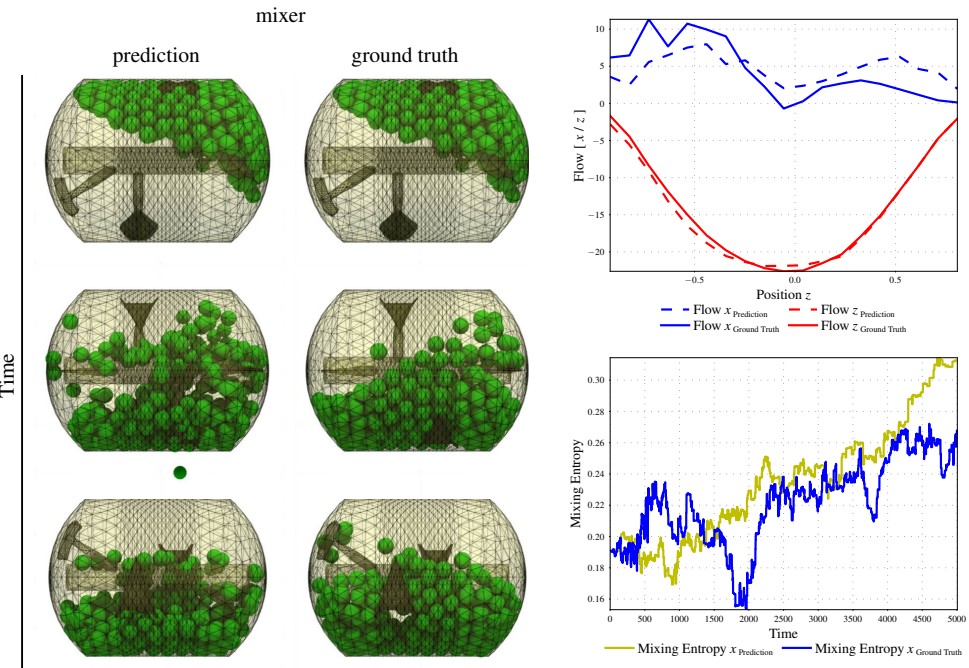

Figure C.2: Mixer dynamics. Left: Particle distributions for particles in a mixer. Simulation data and BGNN predictions are contrasted. Particles are indicated by green spheres, triangular wall areas are yellow, the edges of these triangles are indicated by grey lines. Right: Flow profile (upper right) and entropy plot (lower right) for the used particles. The flow profiles are average velocities of particles at a given z coordinate and are shown for the x (blue)and the z (red) coordinate. The mixing entropies are obtained by splitting particles into two partitions according to a threshold on the respective x coordinate. The prediction entropy curve is shown in yellow while the ground truth simulation entropy curve is shown in blue.

A reason, why we have not been able to successfully train models for dense sampling may be an increased number particle-wall interactions, which might make it more complicated to successfully train such a model, especially if the number of training trajectories is limited.

For unidirectional particle - wall edges, we could successfully learn a model for cohesive particles of the hopper and for both particle types of the rotating drum; the ablation experiment was however not successful for later stages of non-cohesive particles in the hopper. From a methodological point of view it should be noted that the reason why we skipped bidirectional edges was that unidirectional particle-wall interactions are simpler and easier to train. In any case, the ablation experiments did not show advantages over our default model architecture, but seemed to confirm our choice.

Throughout all ablation experiments max-node aggregation did not work well compared to mean aggregation.

Figure C.3: Ablation experiments for cohesive particles within hopper geometries. Different ablations from our default architecture (first three columns, see text) are compared to the ground truth (last column). Particles are indicated by green spheres, triangular wall areas are yellow, the edges of these triangles are indicated by grey lines.

Non-Cohesive Hopper Ablations

Figure C.4: Ablation experiments for non-cohesive particles within hopper geometries. Different ablations from our default architecture (first three columns, see text) are compared to the ground truth (last column). Particles are indicated by green spheres, triangular wall areas are yellow, the edges of these triangles are indicated by grey lines.

Figure C.5: Ablation experiments for cohesive particles within rotating drum geometries. Different ablations from our default architecture (first three columns, see text) are compared to the ground truth (last column). Particles are indicated by green spheres, triangular wall areas are yellow, the edges of these triangles are indicated by grey lines. The circular arrow indicates the rotation direction of the drum.

Non-Cohesive Rotating Drum ablations

Figure C.6: Ablation experiments for non-cohesive particles withing rotating drum geometries. Different ablations from our default architecture (first three columns, see text) are compared to the ground truth (last column). Particles are indicated by green spheres, triangular wall areas are yellow, the edges of these triangles are indicated by grey lines. The circular arrow indicates the rotation direction of the drum.

