# OpenReview forum: "Boundary Graph Neural Networks for 3D Simulations"
_ICLR.cc/2022/Conference — ICLR 2022 Submitted_

### Official Review · Reviewer_bSwm · 2021-10-22

**Correctness:** 4
**Technical Novelty And Significance:** 2
**Empirical Novelty And Significance:** 3
**Recommendation:** 6
**Confidence:** 4

**Main Review:**

The paper is very interesting and well written. Despite its difficulty, it is easy to understand. The background and related work are quite complete and enough to understand the paper. However, there are some parts of the paper that should be explained better to improve its understandability and reproducibility:

1.- The input to the GNN are node coordinates and features but, which kind of features are we talking about?

2.- As commented in point 2, the GNN is composed of 1 encoder, the core message passing layers and the read-out layers. Authors do not specify the configuration of the 5 core-message passing layers nor encoder and read-out layers.

3.- mij appears in (4) and (9) but authors do not explain its meaning. It could be the message passing of Gilmer et al 2017. However, in this case, the mij is a linear combination of the neighbors embedings that have already introduced in both formulas. It would follow the Gilmer proposal if it was eij instead of mij. Please, this part must be explained better.

4.- In page 7, the authors state that "We use 5 message passing layers with 128 and 512 nodes for intermediate node and edge representation". But, what structure does it has?. This is the first time that authors talk about using edges in the GNN model. Which information do the edges have? How are their embedings built?

5.- In the results, the authors comment that they introduce a 25% of variability in the number of particles but do not justify this decision.

6.- A comparison with Sanchez-Gonzalez 2020 would be interesting to show the limits of the improvement introduced.

**Summary Of The Paper:**

This paper presents a novel contribution to the use of Graph Neural Networks (GNN) in the simulation of 3D physical phenomena. This contribution consists of the optimization in the calculation of the interactions of the particles with the contour surfaces that allows to optimize the need to add nodes in these limits. The results show an acceleration of the simulation calculation without affecting the accuracy of the result.

**Summary Of The Review:**

Pros:
1.- A very interesting contribution to GNN applied to 3D simulation.
2.- It could be a first step towards using GNNs in complex mechanical and biomechanical simulation.
3.- The increase in speed without affecting too much the precision is a very interesting contribution.

Cons:
1.- The model must be explained better for easier understanding (check the main review).
2.- It is not clear the differences between the base GNN model (without the dynamic boundary nodes/edges addition) and the one proposed by Sanchez-Gonzalez 2020. (Maybe because cons 1).
3.- Lack the comparison with other similar approximations.

---

> ### Author Response · Authors · 2021-11-22
> **Response to Reviewer bSwm**
>
> We thank the review for the insightful comments and the questions which allowed us to improve the manuscript. In the following we answer the reviewer’s questions.
>
> Q1: The input to the BGNN are the initial particle positions and the particle positions at the 5 previous frames of the simulations. Further inputs are particle type and the coordinates of the triangle mesh. We do now explicitly specify this in appendix (C.2).
>
> Q2: The encoder, decoder and the core message passing layers are simple MLPs with two layers,  ReLU activations after the first layer, and, Layer normalization  after the second layer. We adopted this architecture from the graph network implementation of Battaglia et al 2018. For the core message passing layers, we used residual connections. We describe these implementation details in the appendix (C.2).
>
> Q3/Q4: Thanks for making us aware of this. Indeed, due to page space limitations, we tried to write down our formulas for message passing a bit too compactly. We now updated the corresponding message passing formulas in our manuscript. m_ij describes neural information at edge ij. At the encoding layer, the edge attributes (a_ij) are distances, and distance vectors (or deterministic functions of this information like the inverse of distances) between the corresponding nodes.
>
> Q5: The purpose of this variability study is to allow a comparison of the model’s precision to the effect of a varying problem parameter (in our case the particle number). The extend of this variation is chosen arbitrarily as the required precision in a real-world use case varies from case to case.
>
> Q6: Our methods builds upon Sanchez-Gonzalez 2020 and extends it. A direct application of Sanchez-Gonzalez 2020 to triangularized geometries is not possible. However, we tried to do ablation experiments by densely sampling triangle boundaries in the appendix (C.6).

---

### Official Review · Reviewer_QRXr · 2021-10-24

**Correctness:** 3
**Technical Novelty And Significance:** 2
**Empirical Novelty And Significance:** 2
**Recommendation:** 5
**Confidence:** 4

**Main Review:**

Strengths:

S1: The proposed approach sounds like a very plausible way to encode triangularized boundary surfaces as nodes and edges in a graph.

S2: The idea is very simple, and can be combined in a very straightforward way with existing models, without any detrimental effects to them (e.g. models can still learn a variety of dynamics, generalize to large systems, etc.).

Weaknesses:

W1: The technical contribution of this paper is sort of a "featurization" contribution, rather than a "modelling" contribution, e.g. it focuses on the problem of how to build an input graph that you can give to a GNN for a particle simulation with triangular boundaries. While this is the first paper (that I know off) that focuses on that specifically, some of the previous papers were already using sort of similar strategies. For example, Li et al. were already using a single particle to represent each wall in the container (rather than multiple), also adding edges according to the normal/closest distance between the surface and the particles (which does not present the scalability problems indicated by the authors). Ummenhofer et al. also used normals as boundary node features. Sanchez-Gonzalez et. al, included clipped normal distances to walls as additional node features to account for the cubic container to avoid the need to use large numbers of particles for the container walls. The approach proposed in this paper unifies all of those approaches in a sensible and general way, however, the technical novelty feels perhaps a bit too incremental. For comparison, in those 3 previous baselines, the choice of how to build the input graph is sort of a minor implementation detail compared to other contributions, while in this case, the paper focuses solely on that encoding detail.

W2: The paper does not compare against any of those previous baselines using alternatives ways to encode boundaries (e.g. approaches that model surfaces as dense layers of particles). This is probably fine since the authors are not trying to claim that the model accuracy is better at modeling the dynamics, just that it is more efficient as it can do the same with less edges and nodes for the boundaries (and you don't need to actually run a baseline to believe that less nodes and edges will yield a faster model). However, it does take a bit away from the experimental results from the paper, since it probably indicates many of the previous baselines can also model those domains to similar accuracy, and achieve similar flow, mixing entropy and generalization performance.

W3: Authors claim that the model can be faster than state of the art simulation methods. I believe this is probably true, but I feel it is not very well supported as it is a very small part of the paper. E.g. I am not sure if it is fair to say that BGNNs are non-optimized for GPUs, since neural networks are sort of optimized for GPUs.

Other comments:

O1: The description of the contribution could be a bit more clear. If I understood correctly, the idea is quite simple: For every pair of real particle and boundary triangle that are close, add an extra virtual particle at the closest point of the boundary triangle, including the normal of the boundary as an extra feature. Once you have added those particles, just apply *almost* the same model as in the baselines on teh resulting graph. Currently the description of this takes two sections and almost 3 pages (middle of page 3 to near-top of page 6), and there is quite a bit of redundant info between section 3 and 4. I think the paper could benefit from a simpler and shorter description. Or perhaps I am missing something about the methods, in which case I apologize.

O2: Some of the math seems missing some descriptions and parts. For example I believe m_ij in equation 4 and 9 are not defined anywhere. Also, I believe the paper does not currently indicate which of the node features p_i and x_i are used to build the initial set of node embedded features h_i. I assumed absolute positions x_i are *not* used to compute the embedding and just used to build the relative positional features via subtraction at the edges (so the model is translation equivariant), but I am not sure whether absolute positions are used as features or not is clarified anywhere.

O3: Table 2, it is unclear what the time steps column indicates. At first glance it seems to read that BGNN takes 158s to run 100 timesteps, while LIGGGHTS takes 356s to run 1 timestep. Consider renaming the columns to "timesteps/per step", "step size" or something like that, and "full trajectory wall clock time".

**Summary Of The Paper:**

The paper proposes BGNNs, which are an extension to existing GNNs for particle simulation (e.g. Li et al, Ummenhofer et al, Sanchez-Gonzalez et al), aimed at improving the efficiency of the model when particles are near boundaries with complex triangular geometries.

Specifically, the authors propose to parametrize triangular boundaries such that, for any "real" particle/node close to any triangle of the boundary surface, a new "virtual" particle/node is added at the intersection between the boundary surface and the normal between the real particle and the surface. Furthermore this "virtual" particle/node has additional features such as the normal vector to the surface. The authors then implement the model in two geometries (drum and hopper) for two types of simulation (cohesive and non cohesive particles).

The main claims given the model are:

C1. This approach to describe boundaries is more efficient than those approaches used in prior work.

C2. The model makes predictions for the chosen domains that match quantities of interest to the engineering problem, for each simulation type, and can generalize to larger systems at test time.

C3. The model can be faster than state of the art hardcoded simulation methods.

**Summary Of The Review:**

The paper proposal for how to create an input graph for a particle simulation interacting with complex geometries is plausible. However, considering how large overlap with prior work there is, except for the "graph encoder" part of the model, I am not sure if this contribution on its own is large enough to meet the ICLR bar. Specifically, I am not sure the paper will bring many new insights to the average ICLR attendee, beyond seeing it as an implementation detail for existing models. In this sense I am borderline between weak accept and weak reject.

I would be much more inclined towards a stronger accept if the focus was a bit more applied:
1. Here's an actual application, with realistic data, that engineers really care about, and also here's a specific set of generalization settings that engineers really care about.
2. Here's how you adapt an existing method (e.g. one of Li et al, Ummenhofer et al, Sanchez-Gonzalez et al, Pfaff et al.) to this problem including changes to the encoder to avoid wasting compute on large regions consisting only of boundaries, which ideally really matters if you want to get good running times in that application.
3. Here's a detailed study of why engineers in those fields should prefer the learned model over other state of the art non-learned model. Including plausibility of generating a training dataset of the required size, limits of generalization, running time, trade offs, etc.

Of course, it may be that a different venue could be more appropriate for a paper like that. Although as an ML conference attendee, I would personally be interested on the specifics of any evidence of the promise of end-to-end learned models replacing traditional simulators in any domain.

---

> ### Author Response · Authors · 2021-11-22
> **Response to Reviewer QRXr**
>
> We thank the reviewer for the thoughtful comments. In the following we would like to comment on the three weaknesses named by the reviewer.
>
> W1: Although it’s true that other approaches also considered boundaries for their 3D simulations, these were often quite simple ones (e.g., the containers in Sanchez-Gonzalez et. al or Li et. al). The usage of triangularized boundaries adds an additional level of complexity, which justifies the approach we took. We want to explicitly stress the abundance of such problems in every day’s engineering life. Some approaches use dense sampling to represent boundaries (e.g., Sanchez-Gonzalez et. al for their 2D simulations or Ummenhofer et al.), and we therefore provide ablation experiments towards this direction in App. C.6. The dense sampling approach is much harder to train, and despite extensive hyperparameter tuning we by far could not reproduce our results.
> We agree with the reviewer that we suggested some other interesting novelties (e.g., orientation-independent vector representations) that might for themselves be interesting concepts to the Deep Learning community (unfortunately the corresponding reference to the appendix chapter was missing in the main text, which we added now).
>
> W2: As mentioned above, we refer to App. C.6 which discusses ablation studies that address the relevant related papers.
>
> W3: The reviewer is right that traditional simulations often only employ CPUs for their computations and therefore runtime comparison to Deep Learning based simulations are difficult as traditional simulations often don’t make use of the full compute capabilities, that would be available. However, we think, that this is not only specific to comparisons in our manuscript.
>
> The reviewer also specified three particular points that we should address in order to improve our score:
>
> Concerning the first point, we want to stress that all experiments were not designed for the purpose of highlighting the strengths of BGNNs but are in fact directly motivated by industrial real-world problems regularly faced by simulation engineers in the field. We added a chapter to the appendix (C.4), that studies quantitative effects of particle flows in the context of geometry changes.
>
> Wrt. the second point, we have to point out that our method already relies upon ideas from Sanchez-Gonzalez and tries to extend upon those. In this context Section C.6 in the appendix is relevant, which shows ablations with dense sampling instead of triangular boundaries.
>
> Regarding the third point: The overall goal would be (1) runtime advantages of Deep-Learning based simulations over traditional simulations and (2) differentiable simulations might be useful for further downstream Deep Learning tasks. However, concerning (1) it should be mentioned that this is an active area of research and we don’t yet claim that we have a full demonstration of this in our work, since, as the reviewer correctly mentions, also the creation of training sets and the training time for machine learning models have to be taken into account. For engineers, the selling points might be that a trained BGNN can generalize over different geometries of interest and that it might be more time efficient in the forward pass.
> We claim to be the first paper to address the problem of such complex geometries and also to be the first paper to offer a scalable solution. We hope that many methods will build upon our work.

---

> > ### Comment · Reviewer_QRXr · 2021-11-25
> > **Thanks for the response**
> >
> > Thank you for the response, a couple of clarifications/followups:
> >
> > > W1.
> > Note this comment was not about running ablations (that was W2), because I do believe the approach presented in the paper is sensible and clearly has advantages over dense sampled particles, but about whether the novelty of the contribution was too incremental. This seems to also be one of the main criticisms by other reviewers.
> >
> > > We disagree that building upon prior work limits the novelty. Specifically, Sanchez-Gonzales et al., 2020 do not address the challenges associated with complex geometries in 3D, which is practically and conceptually of unquestionable importance
> >
> > I completely agree with this statement. Building on prior work is always the best option. I think the main criticism here, is that the specific approach of how to build interactions between particles with a triangulated boundary, may not be enough of a contribution for a conference like ICLR, because the solution itself is very simple (which is also a good thing), but also, it is highly technical to a specific applied scenario, e.g. it is more or a featurization contribution than a modelling contribution, which is a bit unusual for a conference like ICLR.
> >
> > > W2.
> > I appreciate the extra experiments. It is indeed very interesting that dense sampling of boundary particles does not work very well, I would have never expected it to work so much worse, just to be much less efficient!
> >
> > > The reviewer also specified three particular points
> > > active area of research and we don’t yet claim that we have a full demonstration of this in our work, since, as the reviewer correctly mentions, also the creation of training sets and the training time for machine learning models have to be taken into account
> >
> > Just want to clarify that the three points were meant to be a sequence, rather than 3 independent points. E.g. whether this type of simulator can replace high-quality commercial simulators is an interesting question, which would be relevant both to the engineering community and the ML community.
> >
> > > For engineers, the selling points might be that a trained BGNN can generalize over different geometries of interest and that it might be more time efficient in the forward pass. We claim to be the first paper to address the problem of such complex geometries and also to be the first paper to offer a scalable solution.
> >
> > Exactly, but I expect that engineers that are machine learning experts trying to solve a problem would already be able to find a solution similar to this without necessarily having to read about it. And engineers that are not machine learning experts would probably benefit more from a paper like this in a more specialized journal.
> >
> > I want to thank the authors again for their responses, and I am looking forward to continue discussing with the other reviewers.

---

### Official Review · Reviewer_kKo7 · 2021-11-02

**Correctness:** 3
**Technical Novelty And Significance:** 2
**Empirical Novelty And Significance:** Not applicable
**Recommendation:** 5
**Confidence:** 3

**Main Review:**

Strengths:

1. The paper proposes a solution to a less studied but important issue. To this end, it is potentially of broad interest in the community.

2. The BGNN seems efficient in that every time it just insert one virtual particle due to its specific mechanism while some other SOTA methods for GNN-based 3D simulation have to calculate/update the positions of several particles.

3. Experimental results demonstrate that BGNN works well on the challenging 3D granular flow processes of hoppers and rotating drums.

Weaknesses:

1. The construction of the BGNN largely follows the framework proposed by Sanchez-Gonzalez et al. (ICML’20), which makes the technical novelty of the work a bit limited.

2. The proposed method is merely tested on two the hoppers and the rotating drums. More diverse geometries should have been considered for experiments to demonstrate the generalisation capability of the BGNN. And there is no comparison with previous methods such as Li et al., 2018, Ummenhofer et al., 2019, Sanchez-Gonzalez et al., 2020 and Pfaff et al.,2020.

3. For Equation 9, the detail of the aggregation is not provided. At least some analysis and ideally an ablation study for a mean or max operation should be presented for this vital step.

In addition, there also exist some minor issues:

1. The title of the paper is inappropriate as the term '3D simulation' is exremely general. Perhaps replace it with '3D granular flow simulation'.

2. Figure 2, as the illustration of one of the key ideas of the paper, is poorly designed and barely informative. It is expected to illustrate how the proposed BGNN dynamically modifies and enhances the graph structure to include boundaries, rather than just exhibits the static local configuration at one time step.


**Summary Of The Paper:**

This paper presents BGNN to dynamically model geometric boundaries for high quality 3D simulation, with the particular focus on the granular flow simulation. For a geometry represented as a 3D surface mesh, the BGNN updates the edges and insert virtual nodes to allow an accurate modeling of particle interactions with triangularised boundaries.

**Summary Of The Review:**

Based on the strengths and weaknesses of the paper listed above, I tend to take a slightly negative stance.

---

> ### Author Response · Authors · 2021-11-22
> **Response to Reviewer kKo7**
>
> We thank the reviewer for the comments and would like to clarify several aspects concerning the weaknesses and the minor issues.
>
> We disagree that building upon prior work limits the novelty. Specifically, Sanchez-Gonzales et al., 2020 do not address the challenges associated with complex geometries in 3D, which is practically and conceptually of unquestionable importance. Similarly, the geometries used in Li et al., 2018 seem to be simple (rectangular walls). An approach that Sanchez-Gonzalez et al., 2020 used for their 2D simulations, and what Ummenhofer et al., 2019 used for their simulations, was, to sample boundaries densely. We therefore employed an ablation experiment towards this direction (App. C.6), in which despite expensive parameter search we were not able to obtain results close to our new approach.
> Concerning Pfaff et al.,2020, while we appreciate their work, their field of application is different than granular particle flow, and a comparison does, therefore, not make sense. However, we agree with the reviewer that the title might be overly general and we are happy to adjust it accordingly for publication.
>
> Overall, we are the first to combine triangular meshes for particle-based granular flow in the context of GNN-based simulation learning and present a practical solution that relies on several crucial methodological novelties.
> To convince the reviewer of the versatility of our approach we added experiments with a further geometry (App. C.5). The new aspect of this mixer geometry is that it includes a boundary element that moves through the granular medium. Our architecture evidently can also learn such dynamics.
>
> Concerning the aggregation operation, we use mean aggregation and specify this now in the paper. We also carried out the suggested comparison experiment on the aggregation type. As the results in App. C.6 show the max aggregation does not work.
>
> We should have clarified that in Figure 2 red points indicate graph nodes and red lines indicate graph edges. Black, dashed lines and black points indicate edges and nodes that are not yet part of the graph but which will eventually be added depending on the distance d. We added this description in the figure caption. Comparing the plots on the left-hand side and on the right-hand side shows that an edge and a node are added, depicting the dynamic graph modification.
>
> Thanks to the reviewer’s comments we could improve our paper and hope that the scores will be adjusted to reflect its significance and novelty.

---

> > ### Comment · Reviewer_kKo7 · 2021-11-29
> > **Response to the authors**
> >
> > Thanks for the feedback of my review. I agree that Sanchez-Gonzales et al., 2020 do not address the challenges associated with complex geometries in 3D. However, the proposed BGNN does not satisfactorily demonstrate that it addresses such challenges well either. Since I am not an expert in this particular field, I wonder if it is widely accpetable to just evaluate the method on two individual samples. And I believe that a proper comparison with  Sanchez-Gonzalez et al., 2020 should be provided to support the authors' claim about the novelty of the paper.

---

### Official Review · Reviewer_93ce · 2021-11-02

**Correctness:** 3
**Technical Novelty And Significance:** 2
**Empirical Novelty And Significance:** 2
**Recommendation:** 5
**Confidence:** 4

**Main Review:**

Strengths:
- The paper is well written and fairly easy to follow and their is sufficient detail in the supplementary material to suggest that reproducibility is not an issue.
- The simulation results and their presentation (both the discrete element method used as ground truth and the simulation of the boundary graph GNN) are well done.
- With regard to the simulation community this method could have practical value since it doesn't require analytical modeling. How easily it could be adapted to other types of machinery though is not entirely clear. The claim that the heuristic method matches ground truth leaves me feeling a bit uneasy though, given the lack of a theoretical analysis or model. Perhaps this is just intractable for granular flows.

Weaknesses:
- A significant weakness in my opinion is the somewhat incremental methodological contribution. Whereas I can accept the practical value of boundary modifications for the application in mind, the idea of adding virtual nodes to a simulation based on a heuristic ball radius, and the consideration of additional node features (as detailed in Section 3) is modest as a contribution to either a "representation" or the "learning" of it, which is the general requirement of ICLR.
- The message passing to update node embeddings seems to be directly based on past work. The specifics of what these messages contain (the new node features, i.e., beyond triangle inclination) are difficult to tease out. Computational considerations, such as how additional coordinate locations are chosen to minimize distance between boundary points and "real" nodes are not addressed. There's an attempt to illustrate the idea in Fig. 2 but it falls short of an algorithm and lacks a complexity analysis.
- Much of Section 4, which discusses further details of how the graph is made "dynamic" appears to be a coverage of implementation issues. Here the additional features features are listed: 1) real or virtual particle, 2) normal vectors but how these features are related to the physics of granular flow is not discussed at any length.
- I found the title to be a bit misleading. While the paper agues for a dynamic modification of a GNN at its boundary for 3D simulations, the experiments and the particular instantiation appear to be almost entirely for granular flow.

**Summary Of The Paper:**

This paper considers dynamic modifications of a GNN, where nodes are inserted, node features are augmented and edges are modified, near the boundary of a triangulated surface mesh, using heuristics. The main intended application area is that of simulating granular flows, which are a challenge to model analytically. Experimental results are shown, comparing the boundary graph GNN outputs with simulations obtained using a compute heavy discrete element method, for two cases of interest in industrial machine design: hoppers and rotating drums. The experimental results include both qualitative illustrations and animations and quantitative comparisons (mixing entropies and flow positions).

**Summary Of The Review:**

Overall, whereas the ideas are easy to follow and the illustrations are well done and the experimental results demonstrate a match between boundary graph GNN outputs and compute heavy simulations (Figs 4 - 6) the contribution of this paper in terms of new methods or representations or representation learning are modest with regard to where the bar is, in my view, for ICLR. I do not doubt that the method has practical value, and clearly a lot of work went into organizing the experiments, conducting them, and putting together the results. Perhaps this paper would be a better fit to a simulation conference or one with a more engineering oriented focus. The idea to dynamically alter graph structure - and as far as I could tell there are not too many other methodological innovations - is a bit thin with regard to novel content. I'm also puzzled by the choice of title - the choice "3D simulations" appears to be quite general, and it is not adequately supported by the paper's content (the focus on granular flows).

---

> ### Author Response · Authors · 2021-11-22
> **Response to Reviewer 93ce**
>
> Thank you for your review.
>
> We strongly contradict the reviewer regarding incremental methodological changes. As written in the general response, our paper is the first to introduce the problem of triangularized geometric boundaries to the Deep Learning community, which none of the existing methods is able to solve properly. Triangularized boundaries appear in abundance in everyday’s engineering machinery. We also are the first to come forward with an efficient algorithm to extend message passing networks to such triangularized geometric boundaries. It is our chosen inductive bias of particle wall interactions which allows a first successful modeling of such problems. In short, we are the first to introduce a hard but omni-present problem to the Deep Learning community, and also the first to present an efficient and scalable solution.
> We further strongly object to the reviewer's statement that there would be no other significant novelties besides dynamically altering the graph. This statement entirely ignores the importance of e.g. the introduction of the problem itself, the orientation-independent representation of normal vectors, the reduced and efficient particle-boundary interactions, or the introduction of a highly efficient GPU-based calculation of the boundary particle’s locations.
> However, to remove any remaining doubts on the significance of our contribution we conducted additional experiments that objectively demonstrate the indispensability of our novelties (see App. C.5 and answer to Reviewer kKo7). The corresponding conclusion is that none of the previously published methods is able to address complex, real-world geometries and that our method is a door-opener in that respect.
>
> We accept the criticism of our title being overly general and are happy to adjust it accordingly for publication.
>
> To conclude, our contribution is evidently of high significance to the community and we hope that this will be properly accounted for in the revised score

---

> > ### Comment · Reviewer_93ce · 2021-11-29
> > **response to reviewers**
> >
> > I'd like to thank the authors for their response. I think there might just be a difference of opinion here. Indeed they are likely correct that this might be the first practical approach to handle triangulated boundaries for particle simulation (but I point out that the title of the paper is overly general - it talks about 3D simulation) with practical advantages over recent related work. However, claiming that orientation-independent normal vector representations is new is hardly true. Those in the computer vision and computational geometry community have been doing that for years, e.g., this is a standard idea in the old image processing literature, where this is done by constructing an appropriate structure tensor. And there's quite a large literature already on handling triangulated objects in neural models in vision and graphics. I'm willing to raise my original rating on this paper, but I do think still that the ideas offered in this paper are modest with regard to the ICLR bar and that it is somewhat of a niche topic.

---

### Author Response · Authors · 2021-11-22
**General response**

We thank all the reviewers for the constructive feedback. We were pleased to see that a number of reviewers found the paper well written and easy to follow, the work having practical value and broad interest to the community, the results demonstrative and well presented, the background and related work quite complete, and the method itself very plausible and easy to combine with existing methods.
However, we distilled the critique of lacking novelty which we want to object.
Our paper is the first to introduce the problem of triangularized geometric boundaries to the Deep Learning community, which none of the existing methods is able to solve properly. Triangularized boundaries appear in abundance in everyday’s engineering machinery. We also are the first to come forward with an efficient algorithm to extend message passing networks to such triangularized geometric boundaries. It is our chosen inductive bias of particle-wall interactions which allows a first successful modeling of such problems.
In short, we are the first to introduce a hard but omni-present problem to the Deep Learning community, and also the first to present an efficient and scalable solution.
In response to queries, we have added several experiments and additions to our paper draft. We think that all of them are valuable contributions, and we are very thankful for the suggestions.

The most important changes are:
* We improved the caption of Figure 2
* We revised message passing formulas slightly in order to define m_ij
* We added details on the architecture
* We added the mixer as a completely new and challenging problem setting
* We trained and analyzed BGNN performance on the mixer geometry
* We ablated BGNNs on the following three design choices:
  * We ablated dense sampling approaches such as those used in Ummenhofer et al., 2019 and Sanchez-Gonzalez et al., 2020
  * We ablated the choice of particle wall interactions
  * We ablated the aggregation operation in our message passing networks

---

### Decision · Program_Chairs · 2022-01-20

**Decision:**

Reject

**Comment:**

This paper describes means of incorporating boundary conditions into graph neural networks used for simulation.  Assorted techniques dynamically adjust computations near a boundary.  Reviewers agreed this work is well written but disagreed regarding whether the scope of the contribution (and application, which focuses mainly on granular flow despite the title of the paper) merits publication at a top conference like ICLR.

The authors rebut the claims of limited scope strongly, but the experiments in the paper somewhat belie the claims of broad scope.  Some comparisons are also missing to Sanchez-Gonzalez et al. 2020, which appears to be closely related.

The AC agrees that the scope of the work and contribution here have not met the bar for publication.  Reviewer QRXr has some thoughtful suggestions for ways to improve this work in future submissions, or sharing with an audience that can better appreciate the application-oriented contributions might be a reasonable direction to take.